# Land Uses for Pasture and Cacao Cultivation Modify the Odonata Assemblages in Atlantic Forest Areas

**Laís R. Santos** [1,2] and **Marciel E. Rodrigues** [1,2,*]

1   Laboratory of Aquatic Organisms ("LOA"), Department of Biological Sciences, Santa Cruz State University (UESC), Ilheus 45662-900, Bahia, Brazil
2   Graduate Program in Tropical Aquatic Systems ("PPGSAT"), Department of Biological Sciences, Santa Cruz State University (UESC), Ilheus 45662-900, Bahia, Brazil
*   Correspondence: merodrigues@uesc.br

**Abstract:** Tropical forests such as the Atlantic Forest are under constant threats from the impact of human activities, mostly being caused by the loss of native forest areas for other land uses. This study aimed to evaluate the effect of changes in land use for pasture and cacao cultivation on the richness and composition of Odonata assemblages in comparison to native forest areas. We also evaluated the species as possible indicators of these different land uses. In total, 64 streams were sampled in southern Bahia, Brazil. A total of 84 species were recorded. The results indicated that changes in land use modify the richness and composition of Odonata assemblages. Regarding composition, our results indicated a difference among the assemblages in the three land use areas and that the native areas maintain more stable assemblages. According to the indicator species analysis, 13 species were recorded as possible bioindicators for different land uses. Changes in aquatic ecosystems and their surroundings caused by different land uses a select group of different species groups, modifying Odonata diversity among these areas. Notably, land uses that maintain a certain integrity of the environment, as in the case of cacao cultivation, are the best alternatives for conserving Odonata biodiversity in comparison with pasture.

**Keywords:** bioindicators; dragonflies; cacao cultivation; pasture; aquatic ecosystems

## 1. Introduction

Land-use changes for agriculture and livestock grazing along with urbanization have intensified over the years, modifying the dynamics of ecosystems and causing the loss of biodiversity [1–4]. In Brazil, the Atlantic Forest, considered a biodiversity hotspot, has been the most widely affected by changes in land uses, including extensive and disorderly timber extraction [1,4–6]. Therefore, it is critical to understand how the type of land use affects ecosystems and their biodiversity [4,7], especially in aquatic ecosystems [8–10].

Among the different types of land uses in Atlantic Forest areas, farming and livestock grazing stand out the most due to the amount (number and size) of modified areas [4]. However, in some regions, other types of land uses are also important. This is the case of areas used for cacao cultivation in the southern region of Bahia, in which cacao is cultivated in the understory of the forest and part of the native vegetation is maintained. This cultivation system is regionally called *Cabruca* [11–13] and is viewed as a sustainable model of production within the remnant Atlantic Forest areas. However, all of these different land uses in Atlantic Forest fragments make them priority areas for conservation actions due to the high loss and fragility of their remaining ecosystems and local biodiversity [2,14], mainly within aquatic ecosystems [10,15,16].

The changes caused by different land uses in the aquatic ecosystems are numerous and affect the chemical conditions as well as the physical and biological structure. These degrade the water quality of water bodies, favor siltation, alter the hydrological regime, and increase the incidence of light [8,17,18]. These factors lead to alterations in the environmental

conditions and reduce the variability of habitats by homogenization, which reduces the availability of food resources and oviposition sites and increases intra-and interspecific competition [18–21]. Moreover, these effects compromise the ability of these environments to maintain their natural communities, thus causing the local extinction of more sensitive species [7,10,16,22–27]

Since species of the Odonata order depend on aquatic ecosystems, they are widely used in studies evaluating the effects of land-use changes on aquatic ecosystems and their surroundings. The species are also used as surrogates of various groups of aquatic insects [28], especially when assessing the impacts caused by different land uses (agriculture, pasture, and urban development) [21,28–33]. However, there are still few studies that have evaluated these effects within the Atlantic Forest and with different types of land use changes (especially in cacao cultivation areas).

Among dragonflies, both the larvae and adults exhibit morphological, ecophysiological, and behavioral characteristics that are closely related to habitat, such as diet and reproduction, oviposition, flight behavior, dispersal ability, and thermoregulation capacity [34–36]. These characteristics divide species into groups that can reflect the quality and integrity of the ecosystems in which they are found [21,29,35], and also help classify species as forest specialists, open area specialists or habitat generalists [33]. For this reason, they can be used as bioindicators of changes in land use [21,28].

The Odonata species considered forest specialists are extremely dependent on the integrity of aquatic ecosystems and surrounding areas. Therefore, they are sensitive to environmental changes and highly susceptible to local extinction when habitats are modified. Species considered open area specialists belong to a group that is adapted to non-forested environments with high levels of solar incidence. They are generally found in natural open areas and in aquatic environments with intermediate levels of alterations of the surroundings. Lastly, species considered habitat generalists are more tolerant of modifications in natural environments and different levels of human impact. For this reason, they are found in areas with various levels of anthropization, including areas where other species cannot develop [21,33].

This study was designed to evaluate the effects of different land uses (pasture, cacao cultivation, and native forest) on the richness and composition of Odonata species within Atlantic Forest areas. In addition, we aimed to identify the possible existence of species that can be regarded as bioindicators of these land uses. We predicted that the pasture areas would have greater species richness than the *cabruca* and native vegetation areas. Modifications in natural ecosystems allowed habitat generalists and open area specialists to colonize these areas and increase local richness [21,33,37–39]. Regarding composition, the prediction was that different land uses would select different species groups. The composition of native forest and *cabruca* areas is similar since it consists of species classified as forest specialists, while pastures tend to be inhabited by species considered to be habitat generalists or open area specialists [33,38]. Considering bioindicator species, we expected the different land uses to be associated with selected group of species that might be considered bioindicators of the native, cacao cultivation, and pasture areas [39].

## 2. Material and Methods

### 2.1. Study Area

This study was conducted in the southern region of the state of Bahia, in the municipalities of Ilhéus, Una, Uruçuca, Itacaré, Buerarema, São José da Vitória, Porto Seguro, and Santa Cruz Cabrália (Figure 1), located in the Atlantic Forest domain. Regional climate according to Köppen-Geiger is classified as Tropical Forest Climate Af (tropical super humid) with evenly distributed rainfall throughout the year. A total of 64 streams were sampled twice of which 24 were in native forest areas, 17 were in in pasture areas, and 23 were in areas of cacao cultivation. The streams are considered low-order (first to third), with an average width of 2 m.

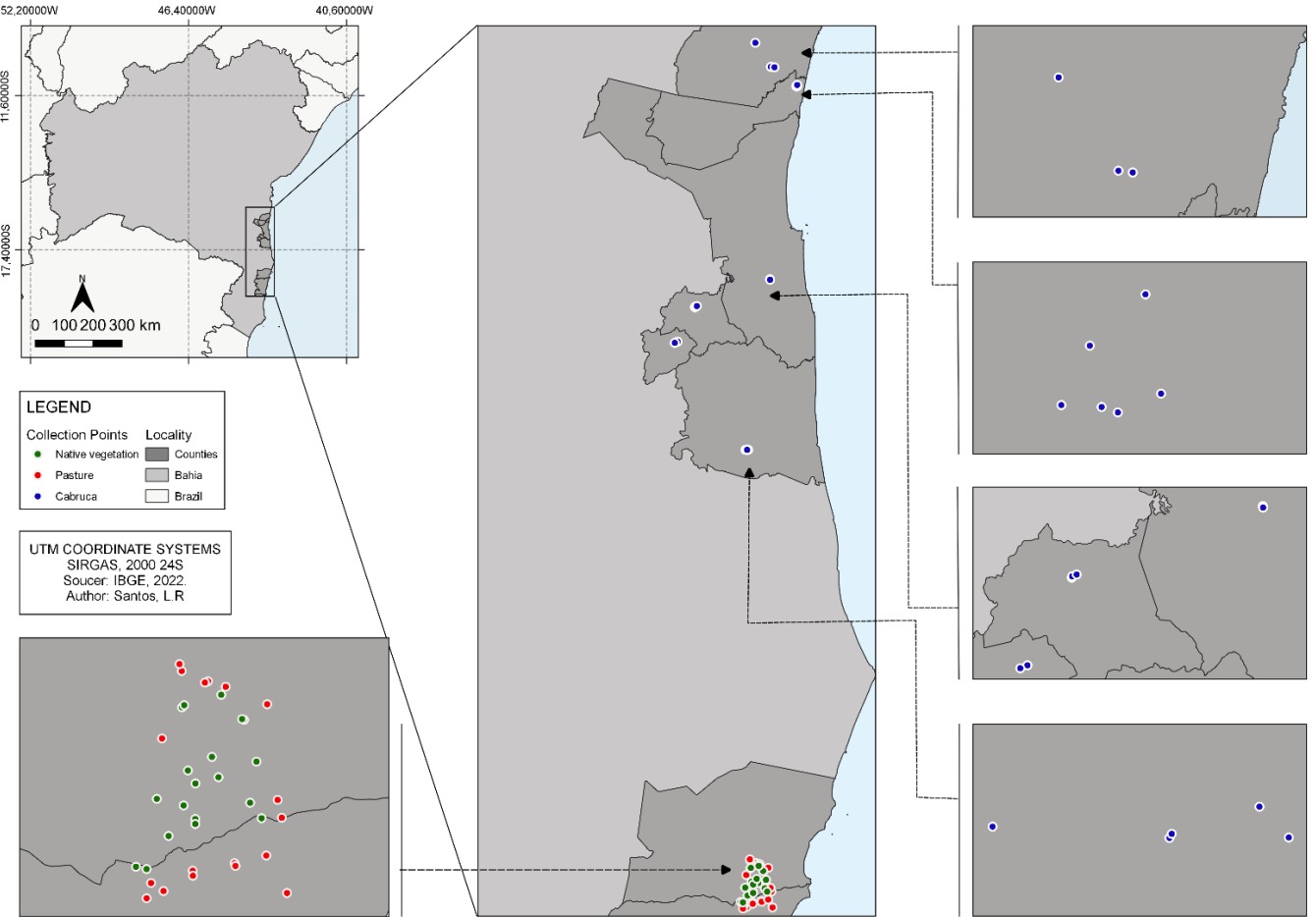

**Figure 1.** Map of the municipalities with the sampling points in areas of native vegetation (green), *cabruca* (blue), and pasture (red).

Collections in native areas were carried out in the municipalities of Porto Seguro and Santa Cruz Cabrália, specifically inside the Estação Veracel Private Natural Heritage Reserve (RPPN), between September and October 2018 and between February and March 2019. In the municipalities of Una and Uruçuca, collections were carried out in permanent preservation areas ("APPs") on private properties between October and November 2019 and July and August 2020. In general, the streams within native vegetation areas have margins with riparian forest and canopy cover, without evidence of physical pollution (disposal of waste discharge of effluents) and with greater physical integrity of the channels (stable margins, little or no evident silting). The streams had widths varying between 78 cm the 501 cm (mean 276 cm and sd 135) and depths varying between 8 cm and 35 cm (mean 28 cm and sd 12).

In pasture areas, collections were carried out in the municipalities of Porto Seguro and Santa Cruz, on private properties and settlements in the region surrounding the Veracel RPPN, during September and October 2018 and in February and March 2019. In general, the streams had little or no riparian vegetation, although some had channel dams to form ponds for drinking water for animals and other purposes on the property. At some points, the margins were unstable and the channel bed was silted. A small amount of household waste was also frequently observed, usually plastic containers. The streams had the width varying between 74 cm and 540 cm (mean 265 cm and sd 128) and depth varying between 12 cm and 63 cm (mean 19 cm and sd 15).

In cabruca areas, collections were carried out on properties of organic cacao producers belonging to the Cabruca Cooperative in the municipalities of Ilhéus, Itacaré, Uruçuca, Buerarema, São José da Vitória and Una, between the months of September and November

2019 and July and August 2020. These sampling sites generally had stable margins with little or no evidence of silting. The riparian vegetation had a slightly closed canopy and many of the sampled areas are used for cacao cultivation until the margins of the streams, forming a canopy over the aquatic ecosystems. The streams had width varying between 31 cm and 292 cm (mean 124 cm and sd 0.70) and depth varying between 5 cm and 96 cm (mean 24 cm and sd 24).

### 2.2. Sampling Method

Adult specimens were sampled within a 100-m section on both margins of the streams. They were collected using an entomological net, with a total sampling effort of 1.30 h for each site. The samples were collected from 9:00 a.m. to 3 p.m., always on sunny days [40]. The collected individuals were sent to the Laboratory of Aquatic Organisms of Santa Cruz State University for identification, which was carried out with the aid of keys [41–45] and other more specific taxonomic keys, along with expert help (see acknowledgments). The collected material is deposited in the collection of aquatic insects of the Santa Cruz State University—UESC.

### 2.3. Data Analysis

To evaluate the response of Odonata richness in the areas with different land uses, the data were tested with a generalized linear method (GLM) using a log-linear [46]. For this analysis, richness was used as a response variable and land uses as predictor variables (native, cabruca, and pasture). To evaluate the relationship of assemblages with the different land uses, principal coordinate analysis (PcoA) and PERMANOVA [47] were performed. For the PcoA, a distance matrix was generated to determine the difference between the assemblages in each land use area and the generated cluster was tested by means of PERMANOVA, from 999 repetitions.

The indicator species analysis (IndVal) for each land use and among the land uses was calculated according to De Cáceres [48]. With this analysis, it is possible to create combinations and evaluate species associated with each type of land use, resulting in association values for each type of use. IndVal also calculates specificity and fidelity. Specificity identifies the probability that a species belongs to a given land use while fidelity indicates how many times the species was recorded for the total sampled points in a particular land use area [48]. All analyses were conducted with the R software and the packages "vegan" [49], "RT4Bio", and "Indicspecies" [50].

## 3. Results

In total, 1558 individuals belonging to 84 species were collected, of which 1217 specimens belong to 34 species were of the suborder Zygoptera and 341 specimens to 50 species were of the suborder Anisoptera. In native forest areas, 514 specimens and 38 species were collected; in the pasture areas, 332 specimens and 43 species were collected; and in the cabruca areas, 712 specimens and 54 species were collected.

Of the 84 species collected, 13 were common to the three types of land uses. A total of 8 species were exclusive in native forest areas, 26 species only occurred in cabruca areas, and 12 were only found in pasture areas. In total, ten species were recorded and found both in the pasture and native forest areas, seven species were found in the native forest and cabruca areas, and eight species were found in both the cabruca and pasture areas (Appendix A).

Species richness among the areas with the different land uses, according to the GLM test, showed a significant difference (AIC = 301.84, df = 61, $p$ = 0.0003), with the model explaining 24% of the data variation, resulting in mean richness of 8.13 for cabruca areas, 4.835 for native areas, and 7.05 for pasture areas (Figure 2)

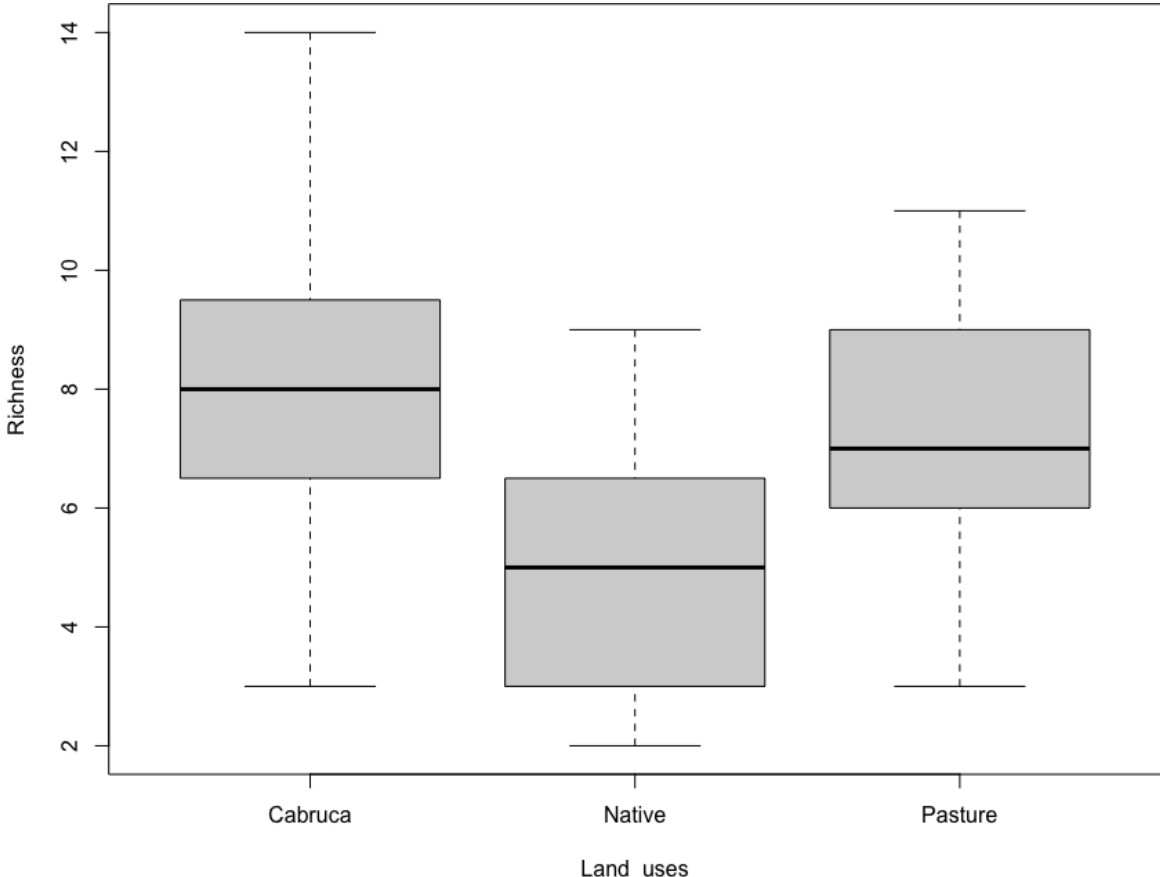

**Figure 2.** Richness of the order Odonata among the different land use areas: cabruca, native, and pasture. The bold line indicates the mean values (8.13 cabruca, 4.835 native, and 7.05 pasture).

Regarding composition, the ordination analysis showed that the assemblages found in streams with different land uses were different from each other (PERMANOVA $p = 0.001$, R = 0.5092). In summary, the species composition differed for each land use (i.e., the assemblages found in the collection sites in each of the land uses were more similar to each other when compared to the different land uses). The assemblages found in native forest areas were most similar to each other, showing a lower variation between the species collected in areas with native vegetation. In the cabruca and pasture areas, the assemblages had the greatest variation in species composition among the sampled sites (Figure 3).

According to the IndVal analysis, some species were identified as possible bioindicators for each of the land uses (Table 1). A total of 20 species were selected. In the cabruca areas, the species were *Acanthagrion aepiolum* Tennessen, 2004, *Argia chapadae* Calvert, 1909, *Aceratobasis nathaliae* Lencioni, 2004, *Epipleoneura metallica* Rácenis, 1955, *Heteragrion consors* Hagen in Selys, 1862, *Erythrodiplax castanea* Burmeister, 1839, and *Perithemis thais* Kirby, 1889. For the native forest areas, the only species was *Heliocharis amazona* Selys, 1853. In the pasture areas, the species were *Ischnura capreolus* Hagen, 1861, *Telebasis corallina* Selys, 1876, *Erythrodiplax paraguayensis* Förster, 1905, *Erythrodiplax leticia* Machado, 1996, *Planiplax phoenicura* Ris, 1912, *Acanthagrion gracile* Rambur, 1842, *Perithemis lais* Perty, 1834 and *Erythemis credula* Hagen, 1861. Moreover, the following indicator species were obtained for area pairs: cabruca-native forest: *Heteragrion aurantiacum* Selys, 1862; cabruca-pasture: *Erythrodiplax fusca* Rambur, 1842; and native forest-pasture: *Argia hasemani* Calvert, 1909, and *Epipleoneura machadoi* Rácenis, 1960. High specificity values, greater than 0.8, were obtained for all species except *Perithemis lais*, with 0.63. These species always occurred in a single type of land use. In contrast, mostly intermediate or low fidelity values were

obtained (almost all of them being less than 0.6). These species had an intermediate or low frequency in the total of each land use.

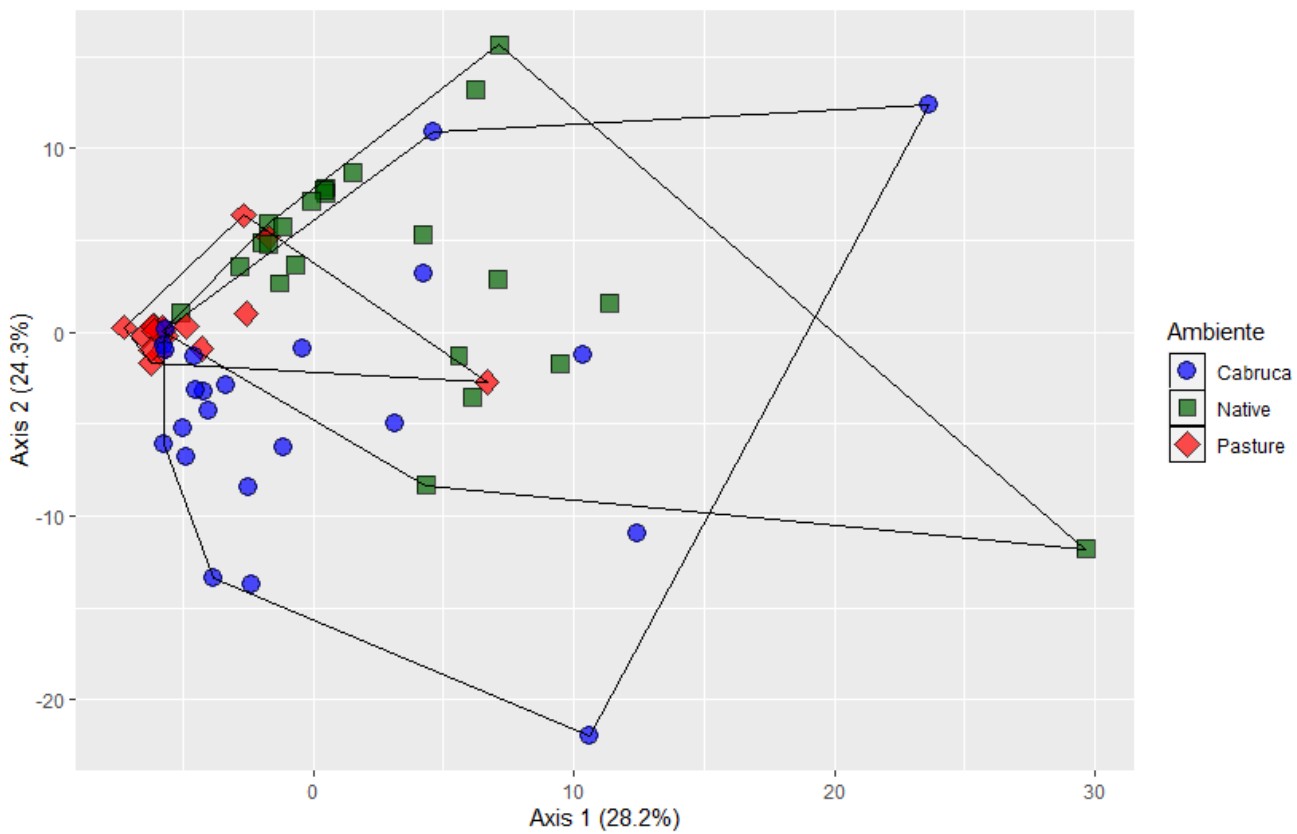

**Figure 3.** PcoA chart demonstrating the similarity of the composition of Odonata assemblages among the land uses: native forest, cabruca, and pasture areas. The dots in green or with the letter N are associated with areas of native vegetation, dots in blue or with the letter C are associated with areas of cacao cultivation, and dots in red or with the letter P are associated with pasture areas.

**Table 1.** Indicator species for the different land uses (cabruca, native forest and pasture) based on the IndVal test.

| Species | Cabruca | Native Forest | Pasture | Index Value | *p*-Value | Specificity (A) | Fidelity (B) |
|---|---|---|---|---|---|---|---|
| *Acanthagrion aepiolum* | x | | | 0.718 | 0.001 | 0.9889 | 0.5217 |
| *Acanthagrion gracile* | | | x | 0.453 | 0.024 | 0.8712 | 0.2353 |
| *Aceratobasis nathaliae* | x | | | 0.417 | 0.017 | 1.000 | 0.1739 |
| *Argia chapadae* | x | | | 0.830 | 0.001 | 0.8339 | 0.8261 |
| *Argia hasemani* | | x | x | 0.733 | 0.001 | 1.000 | 0.5366 |
| *Epipleoneura machadoi* | | x | x | 0.494 | 0.027 | 1.000 | 0.2439 |
| *Epipleoneura metallica* | x | | | 0.417 | 0.011 | 1.000 | 0.1739 |
| *Erythrodiplax castanea* | x | | | 0.417 | 0.024 | 1.000 | 0.1739 |
| *Erythemis credula* | | | x | 0.420 | 0.024 | 1.000 | 0.1765 |
| *Erythrodiplax fusca* | x | | x | 0.712 | 0.003 | 0.9224 | 0.5500 |
| *Erythrodiplax leticia* | | | x | 0.485 | 0.004 | 1.000 | 0.2353 |

**Table 1.** *Cont.*

| Species | Cabruca | Native Forest | Pasture | Index Value | *p*-Value | Specificity (A) | Fidelity (B) |
|---|---|---|---|---|---|---|---|
| *Erythrodiplax paraguayensis* | | | x | 0.531 | 0.002 | 0.9600 | 0.2941 |
| *Heliocharis amazona* | | x | | 0.456 | 0.025 | 1.000 | 0.2083 |
| *Heteragrion aurantiacum* | x | x | | 0.824 | 0.001 | 0.9123 | 0.7447 |
| *Heteragrion consors* | x | | | 0.659 | 0.001 | 1.000 | 0.4348 |
| *Ischnura capreolus* | | | x | 0.737 | 0.001 | 0.9231 | 0.5882 |
| *Perithemis lais* | | | x | 0.432 | 0.0216 | 0.6358 | 0.2941 |
| *Perithemis thais* | x | | | 0.674 | 0.001 | 0.9495 | 0.4783 |
| *Planiplax phoenicura* | | | x | 0.485 | 0.005 | 1.000 | 0.2353 |
| *Telebasis corallina* | | | x | 0.554 | 0.003 | 0.8698 | 0.3529 |

## 4. Discussion

Our results revealed that changes in natural landscapes for other land uses modify the richness and composition of the Odonata assemblages. The cabruca and pasture areas had a greater number of species than the native forest areas. Moreover, the composition differed among the assemblages in the three land uses, which partly corroborates our predictions. Studies assessing the effect of different land uses on Odonata richness have revealed an increase in richness in altered environments when compared to native forest areas [21,33,37]. Modifications in native forest areas cause disturbances of different magnitudes and favor the entry and colonization of Odonata species in these ecosystems. The transformations alter the physical environmental characteristics of the surroundings and aquatic ecosystems and facilitate the colonization of species which are considered open area specialists and habitat generalists [33]. Thus, the different land uses evaluated here (pasture and cabruca) may be maintaining a high richness of species that tolerate slight disturbances when compared to the native forest areas. Of the three evaluated land uses, the cabruca areas exhibited the greatest richness. The cabruca cultivation system caused less severe changes to the ecosystems than the other land uses (namely pasture, agriculture, and urban development). The cabruca areas maintain some of the characteristics found in preserved environments, such as a greater presence of trees, leading to increased canopy cover and, consequently, to greater physical integrity of the channels (stable margins and little or no silting). These characteristics protect aquatic ecosystems from more extensive alterations, while maintaining the physical integrity and quality of water bodies and their surroundings. Thus, the cabruca areas maintain a part of the forest specialist species such as *Forcepcioneura serrabonita*, *Heteragrion consors, and Perilestes fragilis* [51–53] Moreover, some open area specialist species benefit from several of these changes, such as partial canopy opening, especially *Perithemis thais*, *Erythrodiplax paraguayensis* and *Orthemis discolor* [33,54], which increases the richness in these areas compared to native areas.

Although the native areas generally exhibit lower richness than other land use areas, they maintain species that are more sensitive to environmental changes, such as forest specialist species [33]. In our study, the species *Heteragrion aurantiacum, Heliocharis amazona, Leptagrion macrurum, Kiautagrion acutum, Gomphidae* sp1, and *Aceratobasis cornicauda* were only found in the native areas (or exhibited greater abundance in these areas). This finding stresses the importance of native areas to preserving and maintaining the diversity of species that are more sensitive to anthropogenic changes [21,33]. In particular, these environments can maintain highly specific habitats such as phytotelmata, and the loss of these habitats can lead to the local extinction of associated species, such as *K. acutum* and *L. macrurum* recorded in this study. These species are endemic to the Atlantic Forest with few occurrence records [55,56]. Furthermore, *K. acutum* is included in the Red List of Threatened Species as being critically endangered [57].

In terms of composition, more drastic changes were observed in the pasture areas than in the cabruca and native areas. Loss of integrity of the aquatic ecosystem, mainly caused by the removal of vegetation and use of these areas by animals, more abruptly alters the physical structure and quality of these ecosystems and their surroundings and homogenizes the habitats for both the larvae and adults of Odonata [21,33,38]. In this regard, these areas can benefit some species and impair others. Open area specialist species and habitat generalist species that benefit from pastures have been observed in these areas, as in the case of *Ischnura capreolus, Erythrodiplax paraguayensis, Perithemis lais,* and *Erythrodiplax fusca* [32,33,52].

Our results indicated a difference between the assemblages in the three land use areas. The native forest areas showed less variation between the assemblages, while the cabruca and pasture areas differed more from each other. This result reveals that native forest maintains more stable assemblages than other land uses. As they are subject to different levels of anthropic modifications, they also exhibit less similar assemblages. These results have been reported in other studies comparing the composition of Odonata assemblages in native areas with palm trees, pastures, and urban areas [21,33,38].

Composition is a good measure to assess the effect of changes in natural environments on Odonata assemblages. Moreover, it has proved effective in studies such as that of Carvalho [33] for evaluating the effects of extensive palm tree cultivation areas in relation to native and pasture areas in streams of the Cerrado biome [38] and in studies on the impacts of vegetation removal on Odonata assemblages [21,32,37,58]. The different land uses modify the composition of the Odonata assemblages due to changes in the environmental variables of the aquatic ecosystems and their surroundings, which allow species with different ecological and behavioral characteristics to remain and colonize these areas [21,33,35,59].

Previous studies in the Amazon and Atlantic Forest have used a "zygopteran/anisopteran" ratio as an index of anthropogenic effect. Native forests are usually dominated by specialist zygopterans, whereas altered environments with more light contain more anisopterans which lower this ratio [60,61]. Among the species considered as bioindicators, our results revealed the existence of indicator species for each of the three land uses. Among the selected species, almost all had high specificity values (A), thus revealing that species had a high correlation with their respective land uses. However, in relation to fidelity (B), the values were relatively low. The species had low representativeness among the total number of sampled points for each of the land uses. Among the recorded species, *Heliocharis amazona* was classified as an indicator of forest areas. This species is always associated with more pristine environments [37,42] and it is considered a forest specialist [33].

In the cabruca areas, the species *Acanthagrion aepiolum, Aceratobasis nathaliae, Argia chapadae, Epipleoneura metallica, Erythrodiplax castanea, Erythrodiplax fusca, Heteragrion consors,* and *Perithemis thais* were considered possible bioindicators. Notably, some of these species are commonly recorded in more forested areas, as in the case of *Heteragrion aurantiacum, Heteragrion consors,* while others are common in more open or anthropogenic areas, as in the case of *Erythrodiplax fusca* and *Perithemis thais* [21,29,32,33,38]. Furthermore, the cabruca areas help protect more sensitive species, such as those classified as forest specialists, and favors some species considered open area specialists or generalists, thus increasing total richness in these areas.

In the pasture areas, the species *Acanthagrion gracile, Erythemis credula, Erythrodiplax leticia, Erythrodiplax paraguayensis, Ischnura capreolus, Perithemis lais, Planiplax phoenicura,* and *Telebasis corallina* were identified as indicators. In general, these species are found in environments with low canopy cover over the channel, consequently resulting in higher solar incidence, as well as in lentic environments or those with slow flow. Therefore, they are considered open area specialists. Moreover, they are commonly recorded in other studies in natural open areas or areas transformed into pastures [21,29,32,33].

Among the three evaluated land uses, cabruca farming stands out as a sustainable production model within the Atlantic Forest domain in southern Bahia. According to Cassano [62], cabruca agroforestry systems effectively contribute to the conservation of

fauna and flora. Moreover, this contribution is directly related to factors of composition, structure, and management of cacao plantations. Studies carried out with terrestrial invertebrates, birds, and mammals [12,26,63] have shown that cacao-cabruca areas are important for feeding and reproduction and serve as corridors between forest remnants for these species.

In this regard, it is critical to understand how changes in land use affect ecosystems and their biodiversity. Today, changes in land use are highlighted as one of the main anthropogenic problems worldwide, especially in Brazil [4]. Therefore, further knowledge on these effects can support decision-making and proposals for management methods and sustainable use practices to help protect aquatic ecosystems and associated biodiversity [16]. Moreover, ecosystem balance can be maintained without causing interference in local and regional diversity. Thus, production systems that minimize human impacts, as in the case of cabruca systems, are gaining increasing attention. The importance of compliance with current legislation such as the Forest Code [27] should also be stressed. As shown in the present study, riparian vegetation must be maintained in areas of agriculture and pasturing to protect the physical integrity of aquatic ecosystems, their surroundings, and the biodiversity associated with these environments.

**Author Contributions:** Conceptualization, M.E.R. and L.R.S.; methodology, M.E.R. and L.R.S.; formal analysis, M.E.R. and L.R.S.; investigation, M.E.R. and L.R.S.; resources, M.E.R.; data curation, M.E.R. and L.R.S.; writing—original draft preparation, M.E.R. and L.R.S.; writing—review and editing, M.E.R. and L.R.S.; supervision, M.E.R.; project administration, M.E.R.; funding acquisition, M.E.R. All authors have read and agreed to the published version of the manuscript.

**Funding:** This research was funded by Fundação de Amparo à Pesquisa do Estado da Bahia (FAPESB) for fellowships for the first author LRS (process number 073.6787.2020.0007415-43). The Santa Cruz State University—UESC and to the National Council for Scientific and Technological Development—CNPQ, for funding the research project (registered number UESC/PROPP 0220.1100.1693; registered number CNPQ 423737/2018-0).

**Institutional Review Board Statement:** Not applicable.

**Data Availability Statement:** Not applicable.

**Acknowledgments:** We are grateful to the Bahia State Research Foundation (FAPESB) for fellowships for the first author; Santa Cruz State University (UESC) and National Council for Scientific and Technological Development (CNPq); the Graduate Program in Tropical Aquatic Systems (PPGSAT/UESC) and Department of Research and Graduate Studies (PROPP/UESC) for payment of the translation of the manuscript; Estação Veracel RPPN and Cabruca Cooperative for allowing research and for support in field activities and accommodation; and Francisco Valente Neto and Fernando Carvalho for the suggestions and criticisms about this manuscript. To the friends of the Odonatólogos da Bahia group for their support during collections, sorting, material identification and friendship. To the post-graduate program in Tropical Aquatic Systems—PPGSAT/UESC. The Dean of Research and Graduate Studies (PROPP/UESC), for payment of the translation of the manuscript.

**Conflicts of Interest:** The authors declare no conflict of interest. They also state that the funders had no role in the study design; in the collection, analysis or interpretation of data; in writing the manuscript; or in the decision to publish the results.

#### Appendix A. Species Recorded for Different Land Uses in Cabruca, Native Forest, and Pasture Areas in the Sampled Streams of an Atlantic Forest Region in Southern Bahia, Brazil

| SUBORDEM | Family/Species | Abundance | | | |
|---|---|---|---|---|---|
| ZYGOPTERA | CALOPTERYGIDAE | Cabruca | Native Forest | Pasture | Total |
| | *Hetaerina longipes* Hagen in Selys, 1853 | 23 | 25 | 17 | 65 |
| | *Hetaerina rosea* Selys, 1853 | 87 | 113 | 18 | 218 |
| | **COENAGRIONIDAE** | | | | |
| | *Acanthagrion aepiolum* Tennessen, 2004 | 85 | 1 | 0 | 86 |
| | *Acanthagrion cuyabae* Calvert, 1909 | 0 | 0 | 2 | 2 |
| | *Acanthagrion gracile* (Rambur, 1842) | 1 | 0 | 5 | 6 |
| | *Aceratobasis cornicauda* (Calvert, 1909) | 0 | 1 | 0 | 1 |
| | *Aceratobasis macilenta* (Rambur, 1842) | 1 | 0 | 0 | 1 |
| | *Aceratobasis nathaliae* (Lencioni, 2004) | 5 | 0 | 0 | 5 |
| | *Argia chapadae* Calvert, 1909 | 154 | 32 | 0 | 186 |
| | *Argia hasemani* Calvert, 1909 | 0 | 42 | 24 | 66 |
| | *Epipleoneura machadoi* Rácenis, 1960 | 0 | 10 | 14 | 24 |
| | *Epipleoneura metallica* Rácenis, 1955 | 7 | 0 | 0 | 7 |
| | *Forcepsioneura sancta* (Hagen in Selys, 1860) | 1 | 4 | 3 | 8 |
| | *Forcepsioneura serrabonita* Pinto & Kompier, 2018 | 12 | 1 | 0 | 13 |
| | *Idioneura ancilla* Selys, 1860 | 6 | 1 | 4 | 11 |
| | *Ischnura capreolus* (Hagen, 1861) | 4 | 1 | 44 | 49 |
| | *Kiautagrion acutum* Santos, 1961 | 0 | 3 | 0 | 3 |
| | *Leptagrion macrurum* (Burmeister, 1839) | 0 | 10 | 0 | 10 |
| | *Metaleptobasis selysi* Santos, 1956 | 4 | 0 | 0 | 4 |
| | *Neoneura ethela* Williamson, 1917 | 4 | 2 | 0 | 6 |
| | *Neoneura sylvatica* Hagen in Selys, 1886 | 0 | 0 | 5 | 5 |
| | *Nehalennia minuta* (Selys in Sagra, 1857) | 0 | 0 | 4 | 4 |
| | *Telagrion longum* Selys, 1876 | 3 | 1 | 1 | 5 |
| | *Telebasis corollina* (Selys, 1876) | 2 | 7 | 43 | 52 |
| | *Telebasis willinki* Fraser, 1948 | 1 | 0 | 0 | 1 |
| | **DICTERIADIDAE** | | | | |
| | *Heliocharis amazona* Selys, 1853 | 0 | 8 | 0 | 8 |
| | **LESTIDAE** | | | | |
| | *Archilestes exoletus* (Hagen in Selys, 1862) | 4 | 0 | 0 | 4 |
| | *Lestes forficula* Rambur, 1842 | 0 | 0 | 8 | 8 |
| | *Lestes tricolor* Erichson in Schomburgk, 1848 | 0 | 0 | 1 | 1 |
| | **HETERAGRIONIDAE** | | | | |
| | *Heteragrion aurantiacum* Selys, 1862 | 87 | 203 | 20 | 310 |
| | *Heteragrion consors* Hagens in Selys, 1862 | 34 | 0 | 0 | 34 |
| | *Heteragrion gracile* Machado, 2006 | 0 | 2 | 0 | 2 |
| | **PERILESTIDAE** | | | | |
| | *Perilestes fragilis* Hagen in Selys, 1862 | 6 | 4 | 2 | 12 |
| **ANISOPTERA** | **GOMPHIDAE** | | | | |
| | *Gomphoides praevia* St. Quentin, 1967 | 1 | 0 | 0 | 1 |
| | *Gomphoides* sp1 | 0 | 1 | 0 | 1 |
| | *Progomphus* sp | 0 | 1 | 1 | 2 |
| | *Progomphus montanus* Belle, 1973 | 0 | 0 | 2 | 2 |
| | *Phyllogomphoides* sp | 0 | 1 | 1 | 2 |
| | *Zonophora calippus* Selys, 1869 | 0 | 1 | 2 | 3 |
| | **LIBELULIDAE** | | | | |
| | *Anatya guttata* (Erichson in Schomburgk, 1848) | 5 | 0 | 2 | 7 |

| | | | | |
|---|---|---|---|---|
| *Anatya januaria* Ris, 1911 | 2 | 0 | 0 | 2 |
| *Dasythemis essequiba* Ris, 1919 | 1 | 0 | 0 | 1 |
| *Dasythemis venosa* (Burmeister, 1839) | 1 | 0 | 0 | 1 |
| *Diastatops obscura* (Fabricius, 1775) | 3 | 0 | 8 | 11 |
| *Diastatops nigra* Montgomery, 1940 | 9 | 3 | 0 | 12 |
| *Elasmothemis alcebiadesi* (Santos, 1945) | 6 | 0 | 0 | 6 |
| *Elasmothemis cannacrioides* (Calvert, 1906) | 0 | 7 | 3 | 10 |
| *Elga leptostyla* Ris, 1909 | 1 | 0 | 0 | 1 |
| *Erythemis carmelita* Williamson, 1923 | 1 | 0 | 0 | 1 |
| *Erythemis credula* (Hagen, 1861) | 0 | 0 | 4 | 4 |
| *Erythemis vesiculosa* (Fabricius, 1775) | 0 | 1 | 0 | 1 |
| *Erythrodiplax avittata* Borror, 1942 | 0 | 1 | 3 | 4 |
| *Erythrodiplax castanea* (Burmeister, 1839) | 14 | 0 | 0 | 14 |
| *Erythrodiplax famula* (Erichson in Schomburgk, 1848) | 1 | 0 | 0 | 1 |
| *Erythrodiplax funerea* (Hagen, 1861) | 0 | 3 | 0 | 3 |
| *Erythrodiplax fusca* (Rambur, 1842) | 62 | 9 | 30 | 101 |
| *Erythrodiplax latimaculata* Ris, 1911 | 1 | 0 | 0 | 1 |
| *Erythrodiplax leticia* Machado, 1996 | 0 | 0 | 6 | 6 |
| *Erythrodiplax lygaea* Ris, 1911 | 1 | 0 | 1 | 2 |
| *Erythrodiplax maculosa* (Hagen, 1861) | 3 | 0 | 0 | 3 |
| *Erythrodiplax media* Borror, 1942 | 4 | 0 | 0 | 4 |
| *Erythrodiplax paraguayensis* (Förster, 1905) | 0 | 1 | 17 | 18 |
| *Erythrodiplax umbrata* (Linnaeus, 1758) | 5 | 2 | 3 | 10 |
| *Erythrodiplax* sp1 | 1 | 0 | 0 | 1 |
| *Erythrodiplax* sp2 | 2 | 0 | 0 | 2 |
| *Erythrodiplax* sp3 | 1 | 0 | 0 | 1 |
| *Macrothemis tenuis* Hagen, 1868 | 4 | 0 | 0 | 4 |
| *Micrathyria atra* (Martin, 1897) | 0 | 1 | 1 | 2 |
| *Micrathyria artemis* Ris, 1911 | 8 | 0 | 2 | 10 |
| *Micrathyria catenata* Calvert, 1909 | 1 | 0 | 2 | 3 |
| *Micrathyria mengeri* Ris, 1919 | 0 | 0 | 1 | 1 |
| *Micrathyria ungulata* Förster, 1907 | 12 | 0 | 2 | 14 |
| *Nephepeltia phryne* (Perty, 1833) | 1 | 0 | 0 | 1 |
| *Oligoclada abbreviata* (Rambur, 1842) | 1 | 0 | 0 | 1 |
| *Oligoclada umbricola* Borror, 1931 | 1 | 0 | 2 | 3 |
| *Orthemis attenuata* (Erichson in Schomburgk, 1848) | 3 | 4 | 2 | 9 |
| *Orthemis discolor* (Burmeister, 1839) | 4 | 1 | 0 | 5 |
| *Perithemis lais* (Perty, 1833) | 1 | 3 | 5 | 9 |
| *Perithemis thais* Kirby, 1889 | 18 | 1 | 0 | 19 |
| *Planiplax phoenicura* Ris, 1912 | 0 | 0 | 9 | 9 |
| *Tauriphila argo* (Hagen, 1869) | 0 | 0 | 1 | 1 |
| *Uracis infumata* (Rambur, 1842) | 2 | 0 | 0 | 2 |
| *Zenithoptera viola* Ris, 1910 | 0 | 0 | 5 | 5 |
| **Total Abundance** | **712** | **514** | **332** | **1558** |
| **Zygoptera Abundance** | **531** | **471** | **215** | **1217** |
| **Anisoptera Abundance** | **181** | **43** | **117** | **341** |

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
