# Peer review of "Land Uses for Pasture and Cacao Cultivation Modify the Odonata Assemblages in Atlantic Forest Areas"

_diversity, doi:10.3390/d14080672_

Round 1
Reviewer 1 Report
Land uses for pasture and cacao cultivation modify the Odonata assemblages in Atlantic Forest areas by Santos and Rodrigues.
The goal of this study was to describe the adult odonate fauna in native forest, cacao plantations, and pasture in the Atlantic Forest biome of Eastern Brazil—one of the most imperiled rain forest habitats on Earth. The introduction effectively describes the utility of odonates in describing anthropogenic effects on habitats, as their abundance and diversity reflect the ecological integrity of terrestrial and aquatic components of the local ecosystem.
The authors sample over 50 sites in the Atlantic Forest biome of Brazil and describe differences in the odonate communities. The streams are very small first order streams, and each site was only sampled for a total of 1.5 hrs. As such, it is unlikely that these samples effectively captured all of the species that use these sites. Nevertheless, the authors collected more than 1000 individuals representing over 80 species of adult odonates. As such the sampling effort, although neither exhaustive nor a complete census, could effectively describe differences between these habitats.
The analyses were appropriate and complete, and differences in species richness and community composition.
The discussion explains these patterns well, in light of the literature. Native forests harbor fewer species than the impacted habitats, and many of these species are forest specialists that are lost from open pasture habitats. Pastures harbor more species than native forests because, although they lose the forest specialists, they gain more species of open habitat specialists—notably many anisopterans that are dependent on a high light environment to thermoregulate. Interestingly, the cacao plantations sustain the most diverse odonate communities, largely by maintaining more of the forest specialists while also gaining many species that favor disturbed areas and high light environments. This research is important for land management plans in the area, because it demonstrates that maintaining a canopy of native species to shade the cacao also benefits the rest of the forest community and can maintain some of the ecological integrity of the community.
I have made many grammatical suggestions in the accompanying pdf. The only substantive suggestions that I have are the following:
- First, many studies in the Amazon have shown that different forest streams harbor odonate communities that are MORE different from one another than streams in human-altered landscapes. Speciation in different drainages may have led to greater beta-diversity. In altered habitats, the greater environmental stress selects for a subgroup of the same tolerant species, and thus communities in altered areas are usually more similar to one another. This contrasts with this study, where samples from native forests were more similar. I think this is an interesting difference that should be addressed.
- Second, previous studies in the Amazon have used a “zygopteran/anisopteran” ratio as an index of anthropogenic effect. Native forests are usually dominated by specialist zygopterans, whereas altered environments with more light contain more anisopterans that lower this ratio. This would be a nice addition to this study, as well.
Author Response
Response to Reviewer 1 Comments
R: We appreciate the careful review and suggestions in the text.
I have made many grammatical suggestions in the accompanying pdf.
R: We corrected in the text the grammatical suggestions.
Speciation in different drainages may have led to greater beta-diversity. In altered habitats, the greater environmental stress selects for a subgroup of the same tolerant species, and thus communities in altered areas are usually more similar to one another. This contrasts with this study, where samples from native forests were more similar. I think this is an interesting difference that should be addressed.
R: We do not add this suggestion made by the reviewer because we think that the comparison between the studies is not adequate. The Amazon watersheds are very large in length and width, which facilitates the speciation process, especially for Zygoptera. This is not the case for the watersheds in this study.
We also did not analyze beta diversity in situ. And other studies that evaluate different land uses in different biomes have shown that species composition is more similar in native areas when compared to other land uses. As written in the paragraph below in the discussion of our study.
line 267: Our results indicated a difference between the assemblages in the three land use areas. The native forest areas showed less variation between the assemblages, while the cabruca and pasture areas differed more from each other. This result reveals that native forest maintains more stable assemblages than other land uses. As they are subject to different levels of anthropic modifications, they also exhibit less similar assemblages. These results have been reported in other studies comparing the composition of Odonata assemblages in native areas with palm trees, pastures and urban areas [21,33,38].
Second, previous studies in the Amazon have used a “zygopteran/anisopteran” ratio as an index of anthropogenic effect.....
R: We add this suggestion made by the reviewer
line 283: Previous studies in the Amazon and Atlantic Forest have used a “zygopteran/anisopteran” ratio as an index of anthropogenic effect. Native forests are usually dominated by specialist zygopterans, whereas altered environments with more light contain more anisopterans that lower this ratio [60,61]........
References add
60. Oliveira-Junior, J.M.B., Juen, L. The Zygoptera/Anisoptera Ratio (Insecta: Odonata): a New Tool for Habitat Alterations Assessment in Amazonian Streams. Neotropical Entomology. 2019, v.48, p.552–560. https://doi.org/10.1007/s13744-019-00672-x
61. Ribeiro, C., Juen, L., Rodrigues, M.E. The Zygoptera/Anisoptera ratio as a tool to assess anthropogenic changes in Atlantic Forest streams. Biodiversity and Conservation. 2021, v.30, p.1315–1329. https://doi.org/10.1007/s10531-021-02143-5
Reviewer 2 Report
Dear authors,
The manuscript is very well structured, presenting relevant results. I suggest a review of taxonomic names. Some species have recently changed their genus and there are some spelling errors.
Other minor suggestions can be found directly in the text.
Best regards

Author Response
Response to Reviewer 2 Comments
We appreciate the careful review and suggestions in the text. All have been incorporated.
We corrected in the text:
All the parts that were without space between the words.
The spelling in the name of the species and the names that were without italics.
We have added a reference that registers K. acutum (before Leptagrion acutum) for the state of Bahia.
56. Ribeiro, C., Santos, L.R., Rodrigues, M.E. New records of the Critically Endangered Leptagrion acutum Santos, 1961 (Odonata, Coenagrionidae) from southern Bahia, Brazil. Check List. 2021, v. 17(1), p. 59. https://doi.org/10.15560/17.1.59
We corrected the attached table (appendix 1).